# Barriers to implementing contingency management at a methadone treatment clinic: A qualitative study at a tertiary hospital in Tanzania

Paul S. Lawala[1], Christopher F. Akiba[2], Damali L. Kabwali[3], Liness A. Ndelwa[4]
Betuna E. Mwamboneke[4], Albino Kalolo[5]

1 Mirembe National Mental Health Hospital, Dodoma, Tanzania, 2 University of North Carolina at Chapel Hill, Chapel Hill, North Carolina, United States of America, 3 Walter Reed program Tanzania, Henry Jackson Foundation Medical Research International, Mbeya, Tanzania, 4 Mbeya Zonal Referral Hospital, Mbeya, Tanzania, 5 Department of Public Health, St Francis University College of Health and Allied Sciences, Morogoro, Tanzania

* alodala12@gmail.com

## Abstract

### Introduction

Contingency management in addiction behaviors has been widely applied in high income settings. Contingency management entails modification of behavior via the control or manipulation of consequences (contingencies) to the behavior. While a need exists for contingency management in low- and middle- income settings, specifically those of the sub-Saharan Africa region, uptake is low relative to high income settings. This study assessed barriers to implementation of contingency management for methadone treatment clients at the outpatient clinic of a tertiary hospital in Tanzania.

### Methods

This study employed a qualitative design and was conducted at Mbeya Zonal Referral Hospital (MZRH). Guided by the consolidated framework of implementation research (CFIR), data were collected from two sources 1) ten purposively selected key informants (health care workers, methadone treatment clients and hospital leaders) who participated in in-depth interviews, and 2) a mini focus group discussion with five participants (two health care providers, two hospital leaders and one leader of methadone treatment clients). We developed semi-structured guides for in-depth interviews and the mini focus group to explore the key barriers. We analyzed the collected data using thematic analysis.

### Results

Reported barriers revolved around the following key themes: lack of awareness and knowledge regarding contingency management, financial constraints to support implementation, trust between clients and health care workers, Health care workers work load, client

**Data availability statement:** All relevant data are within the paper and its Supporting information files.

**Funding:** No authors received awards the funders had no role in study design, data collection and analysis, decision to publish, or preparation of the manuscript. This study was funded by the sub- Saharan Africa Regional Partnership (SHARP) for Mental Health Capacity Building (NIMH grant U19MH113202) nimh.nih.gov Paul Sarea Lawala received the funding from the SHARP program In addition, no any other author received the fund The funders had no role in study design, data collection and analysis, decision to publish, or preparation of the manuscript.

**Competing interests:** Authors have declared no competing interests.

behaviors and clinic culture. Participants mentioned lack of awareness and knowledge more frequently compared to other themes.

## Conclusion

In the context of specialized outpatient care in Tanzania, contingency management faces a variety of barriers. Deliberate efforts to establish and sustain contingency management in these settings require strategies that attend to the identified barriers. If the barriers are overcome, contingency management implementation and sustainment may follow and ultimately improve methadone related health outcome for patients.

## Background

Behavioral interventions have been successfully used to treat a variety of mental and behavioral disorders, especially by reinforcing appropriate behaviors through positive and negative rewards [1–3]. Contingency management, based on operant conditioning theory, posits that behaviors will increase or decrease as a function of immediate and directly associated consequences (reward or punishment, respectively) [4–7]. Contingency management has been applied in various behavioral conditions such as smoking, alcohol and other drug (AOD) use disorders [8,9]. Contingency management, has been successful in methadone treatment of adults with substance use disorder in hospital and community settings in high income countries [2,10]. However while similar populations in low- and middle- income settings may benefit from contingency management, uptake lags behind high income settings [11,12]. Contingency management can be voucher based or prize based. In voucher-based approach, patients are awarded points that accumulate for submission of drug negative urine sample where as in prize-based CM, patients earn a chance to win a prize (monetary, gift cards, and other prizes) with each negative sample [13]. Cost-effective studies have reported that prize based contingency management is more cost-effective than voucher based contingency management [14].

Contingency management improves substance use outcomes through close monitoring of target behaviors, providing positive reinforcement when the target behavior occurs and removing the reinforcer when the target behavior does not occur [15,16]. The outcomes reported in literature include: substance use abstinence, treatment attendance and quality of life [3,8,17,18].

In methadone treatment settings contingencies include increasing or decreasing methadone dose, urine tests results for heroin use and at home methadone administration [3,19]. The use of contingency management in the context of opiate addiction and other substances is highly efficacious and recommended by existing evidence for substance use adjuvant treatment intervention [5,17,20].

In the Tanzanian setting, literature regarding the use of contingency management is scanty, specifically in methadone clinics. However, calls from similar settings within sub- Saharan Africa have been made to improve implementation of contingency management regarding substance use treatment in adult populations [17].

The aim of this study was to explore the barriers related to implementation of contingency management for methadone treatment clients in a high-volume tertiary hospital in Tanzania.

## Methods

### Study setting

This study was conducted at a methadone clinic at the Mbeya Zonal Referral Hospital, a tertiary hospital in the southern highland zone of Tanzania. The hospital serves approximately

8 million people (15% of the Tanzanian population) in a catchment area of six regions; Katavi, Njombe, Rukwa, Ruvuma, Iringa and Mbeya. The clinic started in 2017, and by November 2021, the clinic had enrolled a total of 403 clients. The average daily attendance stands at 300 clients with default rate of 38%. The graduation rate is recorded at 10% and clients time to graduation averages 5 years.

## Study design

This study employed an exploratory qualitative design to collect data from health managers, health care providers and clients at the methadone clinic. We adopted this approach to best clarify barriers, establish priorities for further research, and gather information about practical solutions to overcome barriers. We followed the Consolidated Criteria for Reporting Qualitative Studies (COREQ) (see S1 File). Data collection took place from August 2020 to January 2021.

## Theoretical framework

The consolidated framework of implementation research (CFIR) commonly guides the process of identifying barriers to the implementation of an intervention in practice settings. Using this framework, we mapped contingency management implementation at the methadone clinics onto four of its domains including: intervention characteristics (contingency management), the inner setting (hospital services), outer setting (mental health policies and practices) and the individuals involved (service users, service providers and the hospital leaders). In the Tanzanian context, the consolidated framework for implementation research has been used to explore barriers related to methadone services and similarly fits the purposes of the present study [21].

## Sample and sampling procedures

The sample for this study comprised of individuals with experience in implementing treatment interventions at the methadone clinic. We believed this group to possess salient knowledge regarding implementation of new interventions like contingency management within the clinic setting. Individuals targeted for enrollment included healthcare managers from the methadone clinic, health practitioners who work directly with the patients on methadone treatment, and patients enrolled in methadone services. Participants in the study were recruited purposively by virtue of their positions, roles and experiences relevant to the question: What are the barriers to implementing contingency management at the Mbeya Zonal Reference Hospital's methadone clinic? A total of 10 participants (five methadone clinic workers, one methadone clinic manager, two Hospital leaders and two clients) were recruited to participate in in-depth interviews. In addition, one mini focus group discussion consisting of 5 participants consisting of health care providers, one leader of methadone treatment clients and hospital leader was conducted. We utilized the mini focus group discussion given the modality's suitability for discussing sensitive and personal issues [22].

## Data collection tools and procedures

Data from the participants were collected using in-depth interview and focus group (FGDs) guides (see S2 and S3 File). Development of interview guides was informed by salient CFIR (Consolidated Framework for Implementation Research) domains and constructs with specific reference to previous studies on contingency management [23]. Since we aimed at exploring the same information from different participants (triangulating information given by providers and clients), the questions in the in-depth interviews and FGD guides were almost similar but collecting individual and

collective opinion respectively. The individual in-depth interviews and focus group discussion were conducted by a trained research assistant (see files attached with this manuscript).

Interviews were recorded and transcribed verbatim and then translated from Swahili to English. We analyzed the transcribed materials using thematic analysis [24] and we were assisted by NVivo software (QSR-international) version 12. Data analysis took account of both the dialogue and the interactions that occurred within the group. We applied an investigators triangulation approach [25,26], whereby two independent researchers, the first and last authors carried out analysis(coding, memo writing, naming and defining themes) of the transcribed material and cross checked illustrative quotes to accompany key themes. The analysis relied on several steps including; 1) familiarization with recordings and field notes; 2) labelling and coding key information; 3) categorizing coded data 4) formulating themes; and 5) interpreting the themes. These steps are summarized in Table 1 which further describes the processes of thematic generation.

## Ethics approval and consent to participate

All study procedures were approved by the ethical review boards at Mbeya Zonal Referral Hospital (Ref No. SZEC-2439). The Mbeya Zonal Referral Hospital IRB works with Zonal center of the National Institute for Medical Research, Mbeya Medical Research Center (MMRC). All participants signed a written informed consent form.

## Results

### Characteristics of participants

Ten participants completed individual interviews, five of which were male and five were female. Participants tended to be middle career with only one younger than thirty years. Five of the participants were health care providers working directly with clients on day- to- day basis. Two were clients receiving services at the methadone clinic, two were the hospital leaders and one served in a management role in a methadone clinic.

Participants in the focus group discussion included 2 health care providers, 2 Hospital leaders and one leader of methadone clinic clients. Of the 5 focus group participants, 3 were males and 2 were females. Three of the participants were less than forty years.

**Table 1. Summary of the data analysis steps.**

| S/N | Steps | Processes |
|-----|-------|-----------|
| 1 | Familiarization with recordings and field notes | • Listening to audio to immerse oneself in the data<br>• Reading and re-reading the transcripts and field notes<br>• Reflecting on the elements of stories or narratives in the transcripts<br>• Going through the transcripts iteratively to understand elements in the story or narrative<br>• Identifying the context of the elements in the stories or narratives |
| 2 | Labelling and coding key information | • Identifying several passages of the text that share same concept<br>• Grouping together similar types of passages of text with same concepts<br>• Developing initial set of codes<br>• Conducting peer debriefing and consensus on the codes |
| 3 | Categorizing coded data | • Creating code categories<br>• Diagramming to make sense of a theme to be developed |
| 4 | Formulating themes | • Sorting and collating all potential categories of codes to themes<br>• Defining and naming the themes by also identifying stories or narratives of each of the themes |
| 5 | Interpreting the themes | • Interpret the developed themes to reflect the logic and relevant account of the theme in relation also to the research question/ objective |

### Key themes on barriers to implementation of contingence management

Thematic analysis of the transcripts revealed six key themes related to barriers for implementing contingency management in the methadone clinic. The identified themes include:

- Lack of awareness and knowledge of contingency management

- Financial barriers to implementing contingency management

- Trust in providing contingency management

- Workload to health care workers

- Methadone treatment clients' behaviors

- Clinic culture.

Furthermore, see Fig 1 for summary of the key themes.

Table 2 provides details of the themes, subthemes and supporting quotes, which are further described below.

### Lack of awareness and knowledge of contingency management

Given that contingency management would be new to the methadone clinic, participants described lack of awareness and knowledge as a key barrier regarding the implementation of the intervention. In addition, hospital staff described how this lack of knowledge hindered their ability to plan or set priorities at the hospital.

> "…. the first thing I can say is understanding, in other words, there is no awareness on contingency management among the staff, from the time we started providing services to methadone clients. That means we do not take this issue seriously, but, it does not mean that we do not see it, we see the need, but we have some sort of heaviness on this issue" (IDI, Health care worker, #2)

While health care workers understood that incentives generally work in improving adherence to clinic visits and medication, most indicated lack of awareness on the use of incentives in the context of structured intervention.

> "Yes, let us say that we are not aware, but once awareness is created, we should also know that incentives can motivate clients. For example, at our clinic we could identify clients who are doing well, that is, they are no longer using substances of abuse [and all parameters we measure are clear], let us say we give them some gifts or whatever, such as a T-shirt and tell them that they are doing well. This will be a memorable event to this client and when another see, will do well also "(IDI, Health care worker, #3)

Narratives from methadone treatment clients revolved around a lack of information on how incentives focused intervention might work, but shared some positive clinical experiences regarding client incentives. Participants welcomed the idea of providing the incentives in a structured way as it could improve client retention in methadone services.

> "Yes, let us say that we are not aware, but once awareness is created, we should also know that incentives can motivate clients. For example, at our clinic we could identify clients who are doing well, that is, they are no longer using substances of abuse [and all parameters we measure are clear], let us say we give them some gifts or whatever, such as a T-shirt

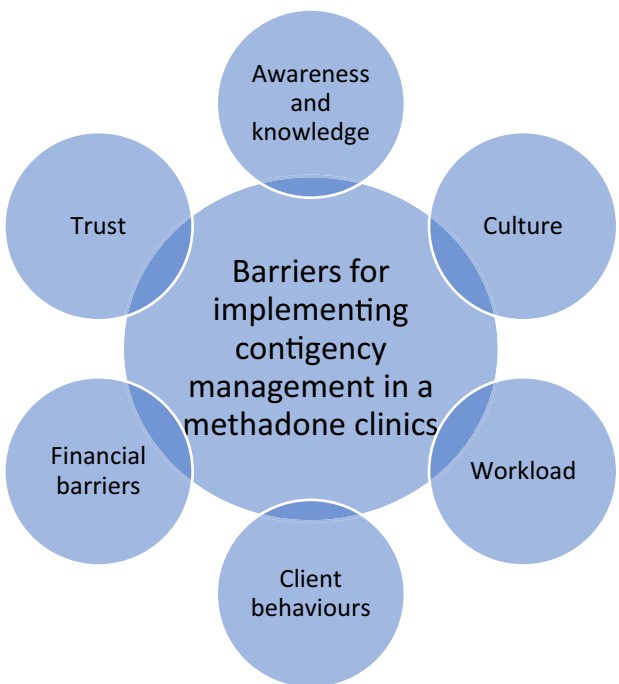

**Fig 1. Schematic diagram of the key themes.**

*and tell them that they are doing well. This will be a memorable even to this client and when another see, will do well also"* (IDI, Health care worker,#3)

## Financial barriers to implementing contingency management

Financing the incentive packages for contingency management in the methadone clinic was one of the important issues reported by participants. This represents an important barrier that could hinder implementation activities even when the initial barriers of is awareness and knowledge are overcome. The key issues related to financing focused on: - inclusion of contingency management activities in existing budgets, priority setting, deciding on a financial incentives package and timely disbursement of funds. Participants revealed that the small hospital budget would hardly accommodate contingency management activities

*"…., I think the big challenge is budget, it is a challenge because we would like to do that [implement contingency management], but when do we do that, we will do it to how many clients?, it will be a challenge, you can do that and it can happen only once, but there is no specific budget may be if a person can do that by himself/herself in the ways he/ she likes "*(IDI, Health care worker,#1)

The participant quotes above describes a fear related to a lack of funds that might hinder contingency management implementation. They note that without adequate funding, providers may be left alone to implement the intervention "in a way they like," reflecting possible challenges related to attaining contingency management fidelity. Some health workers also believed that the hospital administration would not agree to allocate adequate funding for contingency management activities because there are many other unattended priorities amid financial constraints

**Table 2. Themes, sub themes and example of quotes for barriers for implementing contingency management.**

| S/N | Theme | Categories | Types of participants reporting | Supported (example) quotes |
|---|---|---|---|---|
| 1 | Awareness on contingency management activities | Awareness creation to both health care workers and methadone users on the contingency management activities and their related benefits | Hospital leaders<br>Health care workers<br>Clients in Methadone clinic | "… the first thing I can say is understanding, in other words, there is no awareness on contingency management among the staff, from the time we started providing services to methadone clients. That means we do not take this issue seriously, but, it does not mean that we do not see it, we see the need, but we have some sort of heaviness on this issue" (IDI, Health care worker,#2)<br><br>"Yes, let us say that we are not aware, but once awareness is created, we should also know that incentives can motivate clients. For example, at our clinic we could identify clients who are doing well, that is, they are no longer using substances of abuse [and all parameters we measure are clear], let us say we give them some gifts or whatever, such as a T-shirt and tell them that they are doing well. This will be a memorable even to this client and when another see, will do well also" (IDI, Health care worker,#3)<br><br>"But to add on that, he has talked about awareness, but another thing I think we mostly miss is promoting this thing, for example there are things which are given priorities such as family planning, so you can hear even on a radio, therefore, even this one, they can make a certain kind of a promotion like in radio and television" (FDG #3) |
| 2 | Financing the contingency management activities | • Inclusion of CM activities in the budgets<br>• Timely allocation of resources<br>• Priority setting<br>• Deciding on the financial incentives package<br>• Capacity to sustain financial or material incentives | Hospital leaders<br>Health care workers | "The reason and the first thing maybe contingency management activities were not included in the budget". (IDI, Health care worker,#4)<br><br>"…, I think the big challenge is budget, it is a challenge because we would like to do that [implement contingency management], but when do we do that, we will do it to how many clients?, it will be a challenge, you can do that and it can happen only once, but there is no specific budget may be if a person can do that by himself/herself in the ways he/she likes" (IDI, Health care worker,#1)<br><br>"I do not know how the administration will perceive it, it can be a challenge because they can say they have other priorities of other vital things compared to what we are asking for, but if we go to the funding partners may be the challenge will be minor "(IDI, Health care worker,#4)<br><br>"… the challenges which may arise from the leaders may be when you ask for budget or fund, and you find that there are challenges and other priority which are bigger than this one, because this one is a incentives that you are doing, but in the institution, there are many things to do with squeezed budget so you can get that barrier despite that it is a good plan but you may not be able to do that (IDI, Health care worker,#1)<br><br>The first thing is like what I said, when you look at our systems of budgets, we prepare there is no package for methadone services, therefore it is also difficult to implement something which has no budget allocated for it.(IDI, Health care worker,#2)<br><br>"The difficulties which will happen, you know, when you start something, you need to have a number; therefore, you can say may be today we have got 200 users, but if you put incentives, 400 will come, therefore your budget will be in trouble, that is, may be if we can conduct a survey to know all users who are ready to start using methadone" (IDI, Health care worker,#5)<br><br>"Ooh, yes, in providing incentives to clients as part of the contingency management, sometimes funds may delay to reach patients and lead to complaints" (IDI, Health care worker,#6)<br><br>"…, according to my views, I think it is due the limited funds whereas the needs are high when compared to the available money resources, and also the hospital has a big number of methadone users and therefore, this can make that motivation to be not enough for those who will need it because they are many" (IDI, Health care worker,#5)<br><br>"Therefore, the government also was supposed to consider and allocate a certain budget which can enhance it, because a center as a center has no generation funds for that, for example even when you try to see even the medicine is a burden to the center and now if you add the motivation issues" (FGD #2) |

*(Continued)*

**Table 2.** (Continued)

| S/N | Theme | Categories | Types of participants reporting | Supported (example) quotes |
|---|---|---|---|---|
|  |  |  |  | "I want also to contribute at this part that; despite the government, in our community also, there are several institutions especially in the respective regions and not necessarily to be central, we have the sponsors who are within the region, it is better also to explain to those people about something like that. We can gather them at any point even at the institution level, we can gather them privately, then they can be explained about the issue, there are several organizations like; Pepsi, Vodacom, Minerals…, we can explain to them, and not only looking at the government alone, while we have other stakeholders" (FDG # 1) "According to my views, we are not using this because we are relying too much on external financial support, we think that it should be the government to do so, but even ourselves as civil servants we can provide motivation to these people because it is not something huge" (FDG # 2) |
| 3 | Trust in provision of incentives | • Suspicion that the money given to clients/ health care workers is not the same as that allocated for the purpose | Health care workers Clients in the Methadone Clinic | "The users… what I see is for themselves how they receive it, there are those who can receive it well but there are others who can see it like ah, they are giving us because of something…, therefore, the aim of the service which they receive might be weakened because they use methadone so that they can be recovered and hence they can take the incentives negatively" (IDI, Health care worker,#2) "… but also, here you may wonder to see that the money reached on time to the staff but to reach to the patients it is a challenge now by itself" (IDI, Methadone client,#2) "The difficulties we may get are from the clients themselves, that is, people might start giving wrong reasons, and excuses, just to justify that they should be incentivized [especially if the support is money for transport], that is if we give transport fare for those who stay away a person may tell lies where is his/her place of residence? you see? "(IDI, Methadone client,#2) "Another thing, I also think for us as I know the way we are there will be complaints among ourselves that there are others who are favored, like a certain favoritism for those who receive incentives "(IDI, Methadone client,#2) "Another thing…may be let us say the money have been sent for that motivation if it will pass through hospital, we have to expect…, but I am not sure they can be reduced for the reasons I do not know; you may expect 10 million and you will be given 6 million". (IDI, Health care worker,#7) "… so, if clients will be given incentives, I guess we can make positive behaviors. Even if a person can be acting but later, we are sure one thing can change because you cannot deceive to come to take methadone at 07:00 am then tomorrow You come at 10:00 (laughs) while you know that there is a certain motivation. So, I strongly believe that it will build a positive behavior". (IDI, Health care worker,#8) |
| 4 | Incentives of health workers | • Health workers have heavy workload • Compensating health workers' time • Health workers should be given incentives in order to support the program | Health care workers | "…, service providers….I do know if…..if the service providers will be given incentives to sustain their motivation or the incentives will be given to methadone users only so that they go back to innocence" (IDI, Health care worker,#9) "The challenge from the institution [it is in the process of solving though], is that, there are several challenges related to incentives to workers, heavy workloads as a result of congested clinics on daily basis" (IDI, administrator,#2) "The challenges will be for the services providers to claim for extra time money including other rights in accordance with the procedures, regulations, as stipulated in the public service in addition to its guidelines" (IDI, Health care worker,#5) IF5 "Also, I think the service providers or health care workers should be getting something "(IDI, Healthcare worker,#9) |

*(Continued)*

**Table 2.** (Continued)

| S/N | Theme | Categories | Types of participants reporting | Supported (example) quotes |
|---|---|---|---|---|
| 5 | Client behaviors | • Difficult clients<br>• Fulfilling agreement between service users and clinic manager<br>• Hatred against hospital leaders/service providers/motivated group<br>• Managing complaints<br>• Handling large numbers of clients<br>• Cheating nature of clients (not honesty) | Health care workers<br>Clients in the Methadone clinic | "Challenges which I encounter, first is to deal with these clients, there are difficulties due to their nature and the environment in which they come from, and again we did not know them, we did not stay with them to know their behaviors, we have met them here, so it becomes a challenge, you are faced with an abusive language, bad language, you are faced with obstacles and you can be beaten sometimes if you are not careful, therefore all those are challenges" (IDI, Healthcare worker,#10)<br>"The difficulties we face, our clients are in hurry, they do not like to be seen by psychologists, doctors…, they believe in taking medication alone". (IDI, Healthcare worker,#12)<br>"First, our client will behave well, when we see that they have behaviors which are not accepted, if it will be used as a motivation that whoever will come wearing sandals will not be given the incentives which is given to others, the following day all of them will come in the acceptable shoes. So, it will increase…it will increase…discipline" (FGD # 4)<br>"When clients get incentives and are actually motivated to attend clinics, we will get challenges of handling large number of clients. For example, when services are delayed because of small number of health care workers, our clients are not tolerant and sometimes they use a language that is not appropriate" (IDI, Healthcare worker,#10) |
| 6 | Culture | • Non existing culture to motivate clients<br>• Suspicion that there will be favoritism | Hospital leaders<br>Health care workers | "Another thing, I think is also because we do not have a culture to motivate our clients or a clear lack of support mechanisms" (IDI, Health care worker #11)<br>"I think our system of guidelines was not well arranged for this purpose right from the beginning". (IDI, Health care worker, #9) IDI,<br>"I think the way how my brother has said, even us we were trying to deceive them in the streets, this is why others have stopped, because we told them that after a few days you will get something, and when they came, they found that nothing is being provided and when they ask, we tell them that they should recover first because "you did not come here for money but for treatment", others came back and said aah!, let us not follow money but the treatment" (FDG # 5)<br>"I also ask for we service providers to change our attitude towards the community, instead of contributing a lot of money for things like wedding, we can put things like these, we can contribute to things like these as we do to things like wedding, because you will see that we are looking for motivation may be to a 300 hundred people we have, if we had 15 million or 30 million here, they can push a lot of things, here is the same amount as collected in a wedding" (FGD # 2) |

*"I do not know how the administration will perceive it, it can be a challenge because they can say they have other priorities of other vital things compared to what we are asking for, but if we go to the funding partners may be the challenge will be minor"* (IDI, Health care worker, #4)

This participant's description relays skepticism that hospital leaders would make funds available for an intervention like contingency management given overall financial strain amid other gaps in service throughout the hospital. Rather than using the existing hospital budget, they suggest securing funding for contingency management from outside sources

## Trust in providing the contingency management

Both health care workers and clients' narratives highlighted trust is an issue regarding contingency management that stemmed from the prior theme related to financial constraints. Methadone clinic clients described two issues related to trust, first sharing a fear related to favoritism regarding which clients, might receive the intervention, and also that health workers cannot be trusted to provide incentives as some may take for themselves or their relatives.

*"…. but also, here you may wonder to see that the money reached on time to the staff but to reach to the patients it is a challenge now by itself"* (IDI, Methadone client, #2)

*"Another thing, I also think for us as I know the way we are there will be complaints among ourselves that there are others who are favored, like a certain favoritism for those who receive incentives"(IDI, Methadone client,#2)*

Health workers shared their own concerns related to trust but at a different level compared to clients. The health care workers described issues related to delays in disbursement of funds from the administration to the clinics that may jeopardize intervention success. Similarly, the clients pointed out their doubts on how accessible and friendly the contingency management implementation can be to fulfil their needs given any available guides during the implementation process.

*"….. yes, in providing incentives to clients as part of the contingency management, sometimes funds may delay to reach patients and lead to complaints"* (IDI, Health care worker,#6)

Health care workers and methadone treatment clients both shared perceptions regarding how the previous barrier of financial constraints could manifest as a lack of trust between clients and the treatment team, further challenging contingency management implementation. Providers shared specific fears related to leaders diverting funds for contingency management to other administrative works. Methadone treatment clients then shared perceptions that providers may provide incentives to favored clients rather than administering intervention resources among all clients.

## Workload for health care workers

Health workers described how high workloads already challenge the provision of current methadone services and noted how additionally providing contingency management could exacerbate an already heavy workload.

*"The challenge from the institution [it is in the process of solving though], is that, there are several challenges related to incentives to workers, heavy workloads as a result of congested clinics on daily basis"* (IDI, administrator, #2)

The advent of outpatient methadone services resulted in a significant workload increase for providers. The lack of human resources again connects to the earlier barrier related to financial strain on the healthcare system. Participants described how methadone services were simply incorporated into existing mental health services because funds for additional staff did not exist.

In addition, participants indicated the need to incentivize health care workers so as to sustain their motivation to attend to clients amid increased workload challenges

> *"…, service providers… I do know if……if the service providers will be given incentives to sustain their motivation or the incentives will be given to methadone users only so that they go back to innocence"* (IDI, Health care worker,#9)

### Methadone treatment client's behaviors

Respondents described methadone treatment clients' behaviors as possible barriers to contingency management implementation. Narratives from health care workers highlighted their perceptions that some clients are challenging to work with.

> *"Challenges which I encounter, first is to deal with these clients, there are difficulties due to their nature and the environment in which they come from, and again we did not know them, we did not stay with them to know their behaviors, we have met them here, so it becomes a challenge, you are faced with an abusive language, bad language, you are faced with obstacles and you can be beaten sometimes if you are not careful, therefore all those are challenges"* (IDI, Healthcare worker,#10)

> *"The difficulties we face, our clients are in hurry, they do not like to be seen by psychologists, doctors…, they believe in taking medication alone".* IDI, Healthcare worker,#12)

Health workers described perceptions that some of their patients could be hard to work with. They noted how this difficulty could be compounded when monetary incentives like transport reimbursement are made available. The latter barrier connects back to the earlier themes of financial strain and trust between clients and providers

### Clinic treatment culture

Incentivizing clients is not part of the routine practice not only in methadone services, but also in other health services in Tanzania. In implementing contingency management one of the barriers would be how to start making it part of the services. Participants reported this as barrier and might challenge some of the implementation activities

> *"Another thing, I think is also because we do not have a culture to motivate our clients or a clear lack of support mechanisms"* (IDI, Health care worker #11)

Some participants alluded to the fact that, even defining the incentive package (monetary or non-monetary) may be something that is contestable due to absence of such culture, coupled with lack of awareness and knowledge

## Discussion

The objective of the current study was to explore the barriers related to implementation of contingency management for methadone treatment clients s in a tertiary hospital in Tanzania.

We found that in addition to a lack of awareness and knowledge regarding incentives-based interventions like contingency management, the resource limited environment in which methadone services are provided in Tanzania not only represents a major barrier on its own, but additionally produces barriers related to the trust between providers and clients, incentives of health care workers, and client behaviors. Our analysis also identified clinic treatment culture as key barrier to contingency management.

Our first barrier regarding lack of awareness and knowledge of contingency management among health care workers could be explained by the fact the intervention has never been implemented in Tanzania. However, recent substance use treatment research from the region concluded that a incentives based intervention predicted better engagement in treatment, and better health outcomes compared to other types of interventions, and called for the expansion of contingency management [17]. Existing evidence has shown that despite its efficacy in substance dependence treatment, contingency management has not been implemented widely due to lack of awareness among psychiatrists and mental health professionals [2,6].

Implementing contingency management to strengthen behavioral changes central to treatment program success will require awareness creation for policy makers and capacity building for frontline health care workers. The desired behaviors that could benefit from contingency management include: adherence to treatment protocols, not using substances other than prescribed methadone after enrollment, and adherence to clinics visits. Consistent with findings from other reports, behavioral controls are suggested through adding the contingency after the target behavior occurs and removing the contingency when the behavior is not performed [3]. The general implication towards implementing contingency management would be improving clients quality of life, reducing defaulting, and improving daily clinic attendance [27]. Findings from other studies recommend training of front-line health care providers and leaders for successful implementation of contingency management. Some of the suggested methods include didactic training, performance feedbacks and external facilitation in monthly or quarterly order [28]

The theme of limited finances emerged as a major barrier to the successful implementation of contingency management. Participants explained how a lack of sufficient funding permeated multiple services levels ranging from insufficient staffing to negatively impacting the patient-provider relationship. Since the advent of methadone clinics more than 10 years ago in the Tanzania mainland, operational costs have been mainly donor funded on a time limited basis, challenging program sustainability. Donor funding represents a common thread running throughout healthcare provision in the region. A recent systematic review of health financing for universal health coverage in sub-Saharan Africa identified external donor funding as common and similarly found it to undermine healthcare sustainability [29]. The same authors suggest that innovative health financing approaches might include increased government spending on healthcare, bolstering tax compliance and increasing revenue collection efficiency could improve the provision of healthcare [29]. A shift in financing methadone services from external donors to funding partnerships between private ventures and the state may help stabilize existing services and provide a platform for additional interventions like contingency management in the future. In high income countries methadone clinics tend to be jointly funded by public and private firms [30]. Medication cost are usually for methadone, additional opioids and antibiotics for treating other conditions [31,32].

Both clients and health care providers showed their doubts on how the program would be implemented in regards to trustfulness. The emphasis that reinforcement might be given through favors by violating the standard operating procedures in a given scenario, requires in-depth investigations before adopting contingency management practices. The literature has reported two main schedules of reinforcement with clear procedures, a fixed schedule in

which reinforcement is delivered every time the desired target behavior is verified [33], and a variable schedule in which the desired target behavior is not necessarily reinforced each time [34].

Health care workers felt they deserved both financial and non-financial incentives to implement contingency management. Poor incentives is a common barrier to implementing new interventions, and is listed among the second most reported health work force problem in developing countries [35]. Existing studies attest to the role of extrinsic motivation for healthcare workers in enhancing implementation of new interventions and report paying for the performance particularly in areas with shortage of staff as one of the ways of motivating the workforce [36]. However, further evidence report of several factors that affect health care workers motivation including individual factors, work contexts, and social cultural factors. In addition to the framework of motivational factors levels, literature has reported health care reforms as other factors affecting motivation [37]. Therefore, in implementing packages that aim at motivating the workforce with the aim of enhancing implementation of the contingency management in the context similar to our study settings, several factors should be considered

Participants also described clinic culture as a barrier to implementing a contingency management intervention. Existing literature describes how several social and ecological factors come into interplay to influence implementation of contingency management. Such factors operate at, individual, interpersonal, organizational and wider community level [38]. In Tanzania, the culture of providing incentives to motivate patients or clients does not exist. Successful implementation of contingency management in the Tanzanian context will likely require tailoring the intervention to facilitate provision of incentives to the methadone treatment clients. This is in line with existing evidence that report the

that community approach on contingency management-based goal setting groups have been reported successfully working [8,31,39]. This approach will likely be supportive in the Tanzanian setting where the recruitment model for clients is based on the community-facility linkage model. In this model, the groups of clients from the communities are formed to enhance joint learning sessions targeting behavioral in pre-set weekly goals [40]. Further evidence on changing cultural barrier towards supporting contingency management implementation suggest an adjunct use of other psychosocial interventions during implementation process [11]

## Methodological consideration

Our study exhibits several strengths including a purposive sample of participants that spans multiple levels including administrators, clinic staff and clients. We feel this group represents those best suited to describe the barriers of implementing contingency management in a methadone clinic at a tertiary hospital. Our study also looks to advance the treatment of substance use disorders in Tanzania given that evidence-based interventions like contingency management are not routinely implemented in practice settings.

This study was limited by several factors. First, we were able to sample participants from a single clinic operating at a tertiary level. Therefore, the findings should be interpreted only within this context. Second, the fact that participants were predominantly from the supply side (administrators and health care workers), we call for further studies that could exclusively interview clients to get the in-depth reality related to lived experiences of clients. Third, despite efforts to maximize trustworthiness and credibility of our study by pretesting the tools and training research assistants on data collection, we cannot exclude the possibility that participants could have offered socially desirable responses. However, we utilized multiple

participant groups as data sources, as well as multiple data collection modalities to mitigate this concern to the greatest possible extent.

## Conclusion and recommendations

This study demonstrates that the implementation of contingency management in a methadone clinic at a tertiary hospital in Tanzania faces several key barriers: lack of awareness and knowledge, financial constraints, trust, motivation to health care workers, client behaviors and culture. Establishing and sustaining contingency management in these settings requires deliberate efforts to remove the barriers while also fostering the factors that may facilitate implementation. Future research should be conducted in more than one facility and use more rigorous study designs or mixed methods to identify the extent to which barriers are shared across sites, as well as factors that could overcome these barriers and facilitate contingency management's successful implementation.

### Contributions to the literature

- Additional knowledge on the barriers of implementing contingency management in LMICs methadone clinics.

- Setting the future direction on the implementation strategies for contingency management in LMICs methadone clinics.

- Contribution to literature on the promoting use of contingency management intervention for methadone treatment clients in resource limited settings.

## Supporting information

**S1 File. COREQ (consolidated criteria for reporting qualitative research) checklist.**
(DOCX)

**S2 File. In-depth interview guide.**
(DOCX)

**S3 File. Focus group discussion guide.**
(DOCX)

**S4 File. Data sets.**
(DOCX)

## Acknowledgments

Thanks to the participants who gave their time making key information available to this study for successful completion. We are grateful to the leadership at Mbeya Zonal Referral Hospital for allowing their staff and leaders supporting all processes from data collection to the end.

## Author contributions

**Conceptualization:** Paul S. Lawala.

**Data curation:** Liness A. Ndelwa, Betuna E. Mwamboneke.

**Formal analysis:** Paul S. Lawala, Christopher F. Akiba, Kalolo A. Kalolo.

**Funding acquisition:** Christopher F. Akiba.

**Investigation:** Paul S. Lawala, Betuna E. Mwamboneke.

**Methodology:** Paul S. Lawala, Christopher F. Akiba, Kalolo A. Kalolo.

**Project administration:** Paul S. Lawala.

**Supervision:** Paul S. Lawala, Christopher F. Akiba, Damali L. Kabwali, Kalolo A. Kalolo.

**Validation:** Paul S. Lawala.

**Writing – original draft:** Paul S. Lawala, Christopher F. Akiba, Liness A. Ndelwa.

**Writing – review & editing:** Paul S. Lawala, Kalolo A. Kalolo.

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
