## [Decision Letter · Decision Letter 0]

3 Jan 2024

PONE-D-23-19499Barriers to implementing Contingency management at a methadone treatment clinic. A qualitative study at a tertiary Hospital in Tanzania.PLOS ONE

Dear Dr. Lawala,

Thank you for submitting your manuscript to PLOS ONE. After careful consideration, we feel that it has merit but does not fully meet PLOS ONE’s publication criteria as it currently stands. Therefore, we invite you to submit a revised version of the manuscript that addresses the points raised during the review process.

We look forward to receiving your revised manuscript.

Kind regards,

Hamidreza Karimi-Sari

Academic Editor

PLOS ONE

3. In this instance it seems there may be acceptable restrictions in place that prevent the public sharing of your minimal data. However, in line with our goal of ensuring long-term data availability to all interested researchers, PLOS’ Data Policy states that authors cannot be the sole named individuals responsible for ensuring data access (http://journals.plos.org/plosone/s/data-availability#loc-acceptable-data-sharing-methods).

Reviewers' comments:

Reviewer's Responses to Questions

**Comments to the Author**

1. Is the manuscript technically sound, and do the data support the conclusions?

Reviewer #1: Yes

Reviewer #2: Partly

2. Has the statistical analysis been performed appropriately and rigorously? 

Reviewer #1: N/A

Reviewer #2: N/A

3. Have the authors made all data underlying the findings in their manuscript fully available?

Reviewer #1: Yes

Reviewer #2: Yes

4. Is the manuscript presented in an intelligible fashion and written in standard English?

Reviewer #1: Yes

Reviewer #2: Yes

5. Review Comments to the Author

Reviewer #1: This manuscript explores barriers to implementing contingency management in a hospital methadone treatment program in Tanzania. This is a generally well-written paper and a very important topic in light of the global opioid crisis. While most comments are minor, one major limitation may be the lack of opinions from methadone treatment participant clients as well as the inclusion of one treatment client in the focus group among program staff and providers. This study would have been much stronger and more valid if either the clients were not included at all or if more clients were included and were offered participation in their own focus group.

Other comments:

Abstract

1. Does “methadone user” means individuals who are being treated at the methadone clinic? This should probably be “methadone treatment client” or something similar. Methadone user seems like someone who uses methadone illicitly.

Background

1. Maybe replace “mental” with “behavioral” in “mental health disorders” (first paragraph) as it is a bit less stigmatizing.

2. Please update some of the refs in paragraph one. For instance, refs 1, 5, and 6 are outdated.

3. Line 59, second paragraph: Contingency management is recommended by who?

4. The introduction needs a clearer discussion of how contingency management is used in opioid treatment (second paragraph of background). Please add from the literature what behaviors may be reinforced through CM as well as examples of the incentives. For instance, monetary, gift cards, and prizes may be offered, but there is very little mention of that in the manuscripts. This would help support the financial barrier finding cited in the manuscript. Also please add some examples of outcomes that have improved from the use of contingency management in methadone treatment from the literature. Some more recent studies would also be important to add.

5. Also in paragraph 2, the citation 7 (“Implementation support for contingency management: preferences of opioid treatment program leaders and staff” may not be the most appropriate reference to support the statement that CM improves substance use outcomes, as this is a qualitative study exploring OTP staff preferences.

6. First line of second paragraph: Instead of saying no references exist, it may be better to say that the literature is sparse or something similar.

Methods

1. Were participants paid for study participation? Also please mention if consent was obtained.

2. Were participants from the focus group different than the participants completing IDIs?

3. Why was one methadone treatment client included in the focus group with the providers? If there were not enough treatment clients to participate in the focus group, that client probably should not have been included. First, the power structure where a patient from a stigmatized group (substance user) sits in the same focus group as hospital providers (who may be directly or indirectly responsible for treating these patients) may create a situation where the client or providers may not feel as though they can provide honest feedback (as mentioned in the manuscript, there are trust issues). Additionally, questions asked of patients regarding certain interventions that may be implemented in a program would be very different than those asked of staff and providers. Having one client in with a group of providers may contaminate feedback on provider/administration barriers. This should be included as a limitation.

4. More information should be added on how interview guides were developed. Related to comment 3, above, where questions different for staff and clients?

Results

1. I am not sure that figure 1 adds anything to the manuscript.

2. Page 7, line 165: the word “if” may be missing from “in a structured way it could improve…”

3. Page 10, line 248: I believe this should be “latter”, not “ladder” barrier.

4. There are several pages where “.” (periods) and other punctuation are missing (many examples on page 10). The authors should carefully examine the manuscript and edit where needed.

5. Page 10, line 242: “fair” should be “fare” (for transport).

6. The word “motivating” is used throughout (e.g., “motivating” clients). It may be more appropriate to use the word “incentivizing” in most places (e.g., page 10, line 251, “motivating clients is not part of the routine practice.”).

7. Page 10, line 253: where it says “reported presented”: it seems like one of these words should be deleted.

Discussion

1. The authors have written a strong discussion section that includes not only a summary of the barriers found during the study, but also potential solutions from the literature.

2. Page 11, line 276: “awareness” maybe should be “lack of awareness”?

3. Page 11, line 278: “motivate” (or similar) may be a better word than “sharpen”.

4. Page 11, line 281: perhaps change “not using drugs other than methadone” to “not using substances other than prescribed methadone”.

5. Page 11, lines 283-284: “after target behaviors performed” is written twice. Is this an error?

6. Page 11, line 286: What is meant by “reducing defaulting” in this context?

7. Page 11, line 289: What is meant by “in monthly or quarterly order”?

8. Page 12, line 305: What are the “additional opioids” that are mentioned as medication costs?

9. Page 12, line 314: that healthcare workers felt they deserved incentives for providing CM is a finding that was not mentioned in the Results section. It should be added there.

10. Page 13 line 330: The sentence that begins “The role of community approach…” and the next two sentences are unclear/awkward as written. Please revise this section and clarify the information that is being communicated. For example, what are the authors meaning by “community-facility linkage”?

Tables

1. Table 2: It seems as though many of the quotes included in the table are the same quotes from the manuscript. It would be good if the authors could add more new quotes to support the themes.

2. Similar to comment 1, it is hard to tell how much representation there is among participants whose quotes are listed. For instance, the table is dominated by quotes from 34 y.o. male HCW and 28 y.o. female HCW, as well as some other examples of this.

3. Page 24, 3rd quote: transport “fair” should be “fare”.

4. Table 2: The first quote listed under theme 2 (financing) is included under the theme “trust” in the manuscript.

5. Page 26, “motivation of health workers” is not listed as a theme in the manuscript. It is listed as “workload” in the MS.

6. Page 26: I do not see anything in the MS Results about compensating health workers’ time or health workers should be given incentives to implement CM.

7. Page 26, first quote: This quote seems incomplete (ends with “or”).

8. Page 26, second quote: identifying the quote as being from someone who is an administrator, 45, male may lead to issues with confidentiality/privacy as it may be identifiable.

9. Page 26: Under client behaviors, what is “supervising agreed criteria”? Also, “time needed for services”, “motivation for service providers”, and “time constraints” do not seem to fit under this theme and were not discussed in the manuscript.

Reviewer #2: Thank you for your invitation to review this qualitative study that used a CFIR framework to explore the implementation barriers related to contingency management for people who use drugs in a tertiary hospital in Tanzania. The paper identified 6 main themes as barriers to implementing contingency management in the Opioid Treatment Program including 1) awareness and knowledge; 2) Financial barriers to implementation; 3)Trust; 4)Workload; 5)Client behaviors; and 6) Clinic culture

Overall, the manuscript is informative and would potentially add to existing knowledge regarding the barriers to implementing contingency management in other developed countries. However, I think the paper would benefit from language editing (For example, I noticed the authors repeatedly use "substance abuse" instead of "substance use" or places with minor grammar/typographical errors).

---

## [Author Response · Author response to Decision Letter 0]

16 Jul 2024

Date: Jan 03 2024 03:04AM

To: "Paul S. Lawala" alodala12@gmail.com

From: "PLOS ONE" plosone@plos.org

Subject: PLOS ONE Decision: Revision required [PONE-D-23-19499]

PONE-D-23-19499

Barriers to implementing Contingency management at a methadone treatment clinic. A qualitative study at a tertiary Hospital in Tanzania.

PLOS ONE

Dear Dr. Lawala,

Thank you for submitting your manuscript to PLOS ONE. After careful consideration, we feel that it has merit but does not fully meet PLOS ONE’s publication criteria as it currently stands. Therefore, we invite you to submit a revised version of the manuscript that addresses the points raised during the review process.

We look forward to receiving your revised manuscript.

Kind regards,

Hamidreza Karimi-Sari

Academic Editor

PLOS ONE

3. In this instance it seems there may be acceptable restrictions in place that prevent the public sharing of your minimal data. However, in line with our goal of ensuring long-term data availability to all interested researchers, PLOS’ Data Policy states that authors cannot be the sole named individuals responsible for ensuring data access (http://journals.plos.org/plosone/s/data-availability#loc-acceptable-data-sharing-methods).

Reviewers' comments:

Reviewer's Responses to Questions

Comments to the Author

1. Is the manuscript technically sound, and do the data support the conclusions?

Reviewer #1: Yes

Reviewer #2: Partly

2. Has the statistical analysis been performed appropriately and rigorously?

Reviewer #1: N/A

Reviewer #2: N/A

3. Have the authors made all data underlying the findings in their manuscript fully available?

Reviewer #1: Yes

Reviewer #2: Yes

4. Is the manuscript presented in an intelligible fashion and written in standard English?

Reviewer #1: Yes

Reviewer #2: Yes

5. Review Comments to the Author

Reviewer #1: This manuscript explores barriers to implementing contingency management in a hospital methadone treatment program in Tanzania. This is a generally well-written paper and a very important topic in light of the global opioid crisis. While most comments are minor, one major limitation may be the lack of opinions from methadone treatment participant clients as well as the inclusion of one treatment client in the focus group among program staff and providers. This study would have been much stronger and more valid if either the clients were not included at all or if more clients were included and were offered participation in their own focus group.

Other comments:

Abstract

1. Does “methadone user” means individuals who are being treated at the methadone clinic? This should probably be “methadone treatment client” or something similar. Methadone user seems like someone who uses methadone illicitly.

Background

1. Maybe replace “mental” with “behavioral” in “mental health disorders” (first paragraph) as it is a bit less stigmatizing.

2. Please update some of the refs in paragraph one. For instance, refs 1, 5, and 6 are outdated.

3. Line 59, second paragraph: Contingency management is recommended by who?

4. The introduction needs a clearer discussion of how contingency management is used in opioid treatment (second paragraph of background). Please add from the literature what behaviors may be reinforced through CM as well as examples of the incentives. For instance, monetary, gift cards, and prizes may be offered, but there is very little mention of that in the manuscripts. This would help support the financial barrier finding cited in the manuscript. Also please add some examples of outcomes that have improved from the use of contingency management in methadone treatment from the literature. Some more recent studies would also be important to add.

5. Also in paragraph 2, the citation 7 (“Implementation support for contingency management: preferences of opioid treatment program leaders and staff” may not be the most appropriate reference to support the statement that CM improves substance use outcomes, as this is a qualitative study exploring OTP staff preferences.

6. First line of second paragraph: Instead of saying no references exist, it may be better to say that the literature is sparse or something similar.

Methods

1. Were participants paid for study participation? Also please mention if consent was obtained.

2. Were participants from the focus group different than the participants completing IDIs?

3. Why was one methadone treatment client included in the focus group with the providers? If there were not enough treatment clients to participate in the focus group, that client probably should not have been included. First, the power structure where a patient from a stigmatized group (substance user) sits in the same focus group as hospital providers (who may be directly or indirectly responsible for treating these patients) may create a situation where the client or providers may not feel as though they can provide honest feedback (as mentioned in the manuscript, there are trust issues). Additionally, questions asked of patients regarding certain interventions that may be implemented in a program would be very different than those asked of staff and providers. Having one client in with a group of providers may contaminate feedback on provider/administration barriers. This should be included as a limitation.

4. More information should be added on how interview guides were developed. Related to comment 3, above, where questions different for staff and clients?

Results

1. I am not sure that figure 1 adds anything to the manuscript.

2. Page 7, line 165: the word “if” may be missing from “in a structured way it could improve…”

3. Page 10, line 248: I believe this should be “latter”, not “ladder” barrier.

4. There are several pages where “.” (periods) and other punctuation are missing (many examples on page 10). The authors should carefully examine the manuscript and edit where needed.

5. Page 10, line 242: “fair” should be “fare” (for transport).

6. The word “motivating” is used throughout (e.g., “motivating” clients). It may be more appropriate to use the word “incentivizing” in most places (e.g., page 10, line 251, “motivating clients is not part of the routine practice.”).

7. Page 10, line 253: where it says “reported presented”: it seems like one of these words should be deleted.

Discussion

1. The authors have written a strong discussion section that includes not only a summary of the barriers found during the study, but also potential solutions from the literature.

2. Page 11, line 276: “awareness” maybe should be “lack of awareness”?

3. Page 11, line 278: “motivate” (or similar) may be a better word than “sharpen”.

4. Page 11, line 281: perhaps change “not using drugs other than methadone” to “not using substances other than prescribed methadone”.

5. Page 11, lines 283-284: “after target behaviors performed” is written twice. Is this an error?

6. Page 11, line 286: What is meant by “reducing defaulting” in this context?

7. Page 11, line 289: What is meant by “in monthly or quarterly order”?

8. Page 12, line 305: What are the “additional opioids” that are mentioned as medication costs?

9. Page 12, line 314: that healthcare workers felt they deserved incentives for providing CM is a finding that was not mentioned in the Results section. It should be added there.

10. Page 13 line 330: The sentence that begins “The role of community approach…” and the next two sentences are unclear/awkward as written. Please revise this section and clarify the information that is being communicated. For example, what are the authors meaning by “community-facility linkage”?

Tables

1. Table 2: It seems as though many of the quotes included in the table are the same quotes from the manuscript. It would be good if the authors could add more new quotes to support the themes.

2. Similar to comment 1, it is hard to tell how much representation there is among participants whose quotes are listed. For instance, the table is dominated by quotes from 34 y.o. male HCW and 28 y.o. female HCW, as well as some other examples of this.

3. Page 24, 3rd quote: transport “fair” should be “fare”.

4. Table 2: The first quote listed under theme 2 (financing) is included under the theme “trust” in the manuscript.

5. Page 26, “motivation of health workers” is not listed as a theme in the manuscript. It is listed as “workload” in the MS.

6. Page 26: I do not see anything in the MS Results about compensating health workers’ time or health workers should be given incentives to implement CM.

7. Page 26, first quote: This quote seems incomplete (ends with “or”).

8. Page 26, second quote: identifying the quote as being from someone who is an administrator, 45, male may lead to issues with confidentiality/privacy as it may be identifiable.

9. Page 26: Under client behaviors, what is “supervising agreed criteria”? Also, “time needed for services”, “motivation for service providers”, and “time constraints” do not seem to fit under this theme and were not discussed in the manuscript.

Reviewer #2: Thank you for your invitation to review this qualitative study that used a CFIR framework to explore the implementation barriers related to contingency management for people who use drugs in a tertiary hospital in Tanzania. The paper identified 6 main themes as barriers to implementing contingency management in the Opioid Treatment Program including 1) awareness and knowledge; 2) Financial barriers to implementation; 3)Trust; 4)Workload; 5)Client behaviors; and 6) Clinic culture

Overall, the manuscript is informative and would potentially add to existing knowledge regarding the barriers to implementing contingency management in other developed countries. However, I think the paper would benefit from language editing (For example, I noticed the authors repeatedly use "substance abuse" instead of "substance use" or places with minor grammar/typographical errors)

---

## [Decision Letter · Decision Letter 1]

23 Aug 2024

PONE-D-23-19499R1Barriers to implementing Contingency management at a methadone treatment clinic. A qualitative study at a tertiary Hospital in Tanzania.PLOS ONE

Dear Dr. Lawala,

Thank you for submitting your manuscript to PLOS ONE. After careful consideration, we feel that it has merit but does not fully meet PLOS ONE’s publication criteria as it currently stands. Therefore, we invite you to submit a revised version of the manuscript that addresses the points raised during the review process.

We look forward to receiving your revised manuscript.

Kind regards,

Hamidreza Karimi-Sari

Academic Editor

PLOS ONE

Reviewers' comments:

Reviewer's Responses to Questions

**Comments to the Author**

1. If the authors have adequately addressed your comments raised in a previous round of review and you feel that this manuscript is now acceptable for publication, you may indicate that here to bypass the “Comments to the Author” section, enter your conflict of interest statement in the “Confidential to Editor” section, and submit your "Accept" recommendation.

Reviewer #1: (No Response)

2. Is the manuscript technically sound, and do the data support the conclusions?

Reviewer #1: Yes

3. Has the statistical analysis been performed appropriately and rigorously? 

Reviewer #1: N/A

4. Have the authors made all data underlying the findings in their manuscript fully available?

Reviewer #1: Yes

5. Is the manuscript presented in an intelligible fashion and written in standard English?

Reviewer #1: No

6. Review Comments to the Author

Reviewer #1: PONE-D-23-19499

Barriers to implementing Contingency management at a methadone treatment clinic. A qualitative study at a tertiary Hospital in Tanzania

This manuscript explores barriers to implementing contingency management in a hospital methadone treatment program in Tanzania. The manuscript is greatly improved with the revisions. Some issues remain:

Abstract

1. “…for methadone use” would probably be better as “for methadone treatment clients.”

2. In conclusion, “to the s identified”, assuming this should be to the barriers identified? Word missing with just an “s”.

Background

1. Line 54: Word missing, should this be “…in voucher-based settings are awarded points…”? “Are” is missing.

2. Paragraphs 2 and 3 could be combined.

3. Line 69: does this reference (Petry, 2011) discuss the lack of literature on contingency management in Tanzanian settings? If not just delete it.

Methods

1. Were participants paid for study participation? Also please mention if consent was obtained. These important things were not added to the revised manuscript.

2. The authors mention in their comments that the one methadone treatment client included in the focus group with the providers was a leader. This status should be mentioned in the manuscript text.

Results

1. I think in several places at the beginning of the quotes, words of expression such as “aah” and “ooh” and “Eeeheee” could be deleted.

2. There are still several places where “.” (periods) and other punctuation are missing. The authors should carefully examine the manuscript and edit where needed. This was not corrected in the revision.

3. Page 7, quote beginning on line 161: This quote does not support how lack of knowledge hindered leaders’ ability to plan or set priorities. Please either change the quote or add text that shows how the quote supports this sentence.

4. Page 7, is there a quote to support the sentence “Participants welcomed the idea of providing the incentives…”?

5. Line 176: Here, and many places, there are some stray letters, in this case the letter “s”. Not sure if this is an issue with the manuscript upload? In any case, the authors need to carefully review the MS for random issues such as these as well as punctuation errors and typos. More examples on page 9 lines 227-233.

6. Page 10, line 246: word missing? Should this be “attend to clients” instead of “attend clients”?

7. The theme of “methadone client’s behaviors” creating barriers to contingency management is not supported by the quotes. The first quote just discusses the behaviors but is not at all connected to their thoughts on how this is a barrier to CM. The second quote is more to do with the clinic budget and capabilities than clients. Either add in supporting quotes or delete this section/theme. Also, the sentence beginning “The latter barrier…” seems to not connect to the rest of the section.

Discussion

1. Page 12, line 296: now says “Implementing contingency management to sharpen strengthen.” Please fix.

2. Page 13, line 335: This reference is 18 years old and so may not apply here. You could either update the reference or just delete the second part of the sentence (not needed).

3. Page 13, lines 337-340. This is quite confusing as written. It is not clear what the authors are trying to say here. Sentences/language needs to be modified, and authors may need to expand.

4. Page 13, line 346: What is meant by “attend the culture”?

5. The entire paragraph from lines 341-353 needs to be re-read by someone with English as a first language. There are many things that may have gotten lost in translation here (see #4, above, as an example) as well as many missing words. For instance, “in Tanzanian setting” should be “in the Tanzanian setting” as one example.

Tables

1. Table 2: Many of the quotes included in the table are the same quotes from the manuscript. The authors should have different quotes in the table—we have already seen the ones that are listed in the MS. There should be several quotes to support the themes. This was not modified in the revision.

2. Please fix the spacing between quotes in the table. It is difficult to see the quotes separately.

3. Page 27: Under client behaviors, “time needed for services”, “incentives for service providers”, and “time constraints” do not seem to fit under this theme and were not discussed in the manuscript. Actually, only “difficult clients” was discussed in the MS, nothing else that is in this section of the table was in the MS. And, again, most of these quotes do not support the theme of clients’ behaviors as barriers to contingency management. For example, the first quote just discusses that clients may not behave well, but it seems as though that is already and issue and this is not tied into the addition of contingency management. The second quote is a facilitator to implementing CM, it does not mention in any way the poor behaviors of clients as a barrier to implementing CM. Either the theme “client behaviors” needs to be deleted, or quotes that actually support this theme need to be added.

4. In this same theme in the table, what is “wearing sanders”?

5. Under “culture”, why is “suspicion that there will be favoritism” here? It is not in the MS under this theme.

6. The authors need to carefully inspect the tables to make sure that the themes in the tables are expressed the same way as they are in the MS and that the quotes support this.

Overall

Before publishing, the authors need to:

1) Carefully re-read the manuscript and look for grammar and punctuation issues/errors, language and spelling errors, stray letters, etc. The manuscript does not read at all like a final manuscript as is. It reads like a draft.

2) Have someone read through the manuscript whose first language is English, as there are some phrases that are not clearly translated and some sentences that read awkwardly.

7. PLOS authors have the option to publish the peer review history of their article (what does this mean? ). If published, this will include your full peer review and any attached files.

**Do you want your identity to be public for this peer review?** For information about this choice, including consent withdrawal, please see our Privacy Policy .

Reviewer #1: No

---

## [Author Response · Author response to Decision Letter 1]

1 Nov 2024

Dear Editor,

We have responded to the comments raised by reviewers as explained in the cover letter but also the point by point responses in the attached Response to review comments document

---

## [Editor Report · Decision Letter 2]

6 Nov 2024

Barriers to implementing Contingency management at a methadone treatment clinic. A qualitative study at a tertiary Hospital in Tanzania.

PONE-D-23-19499R2

Dear Dr. Lawala,

We’re pleased to inform you that your manuscript has been judged scientifically suitable for publication and will be formally accepted for publication once it meets all outstanding technical requirements.

Kind regards,

Hamidreza Karimi-Sari

Academic Editor

PLOS ONE
---

## [Editor Report · Acceptance letter]

PONE-D-23-19499R2

PLOS ONE

Dear Dr. Lawala,

I'm pleased to inform you that your manuscript has been deemed suitable for publication in PLOS ONE. Congratulations! Your manuscript is now being handed over to our production team.

Kind regards,

on behalf of

Hamidreza Karimi-Sari

Academic Editor

PLOS ONE